# A Smartphone Application for Personalized Tooth Shade Determination

**DOI:** 10.3390/diagnostics13111969

**Published:** 2023-06-05

**Authors:** Tomoya Kusayanagi, Sota Maegawa, Shuya Terauchi, Wataru Hashimoto, Shohei Kaneda

**Affiliations:** Mechanical Engineering Program, Graduate School of Engineering, Kogakuin University, 1-24-2 Nishishinjuku, Shinjuku-ku, Tokyo 163-8677, Japan; am21019@g.kogakuin.jp (T.K.); am21051@g.kogakuin.jp (S.M.); am22044@g.kogakuin.jp (S.T.); am22049@g.kogakuin.jp (W.H.)

**Keywords:** personalized tooth shade determination at home, pTShaDe, relative quantitative tooth shade determination method, tooth whitening, smartphone, smartphone application, mobile health, esthetic dentistry, tooth color measurement

## Abstract

Tooth shade determination methods for evaluating the effectiveness of whitening products at home are limited. In this study, an iPhone app for personalized tooth shade determination was developed. While capturing dental photographs in selfie mode before and after whitening, the app can maintain consistent illumination and tooth appearance conditions that affect tooth color measurement. An ambient light sensor was used to standardize the illumination conditions. To maintain consistent tooth appearance conditions determined by appropriately opening the mouth, facial landmark detection, an artificial intelligence technique that estimates key face parts and outlines, was used. The effectiveness of the app in ensuring uniform tooth appearance was investigated through color measurements of the upper incisors of seven participants via photographs captured in succession. The coefficients of variation for incisors *L**, *a**, and *b** were less than 0.0256 (95% CI, 0.0173–0.0338), 0.2748 (0.1596–0.3899), and 0.1053 (0.0078–0.2028), respectively. To examine the feasibility of the app for tooth shade determination, gel whitening after pseudo-staining by coffee and grape juice was performed. Consequently, whitening results were evaluated by monitoring the *∆E_ab_* color difference values (1.3 unit minimum). Although tooth shade determination remains a relative quantification method, the proposed method can support evidence-based selection of whitening products.

## 1. Introduction

Despite an increasing demand for whiter teeth and numerous products available for teeth whitening at home [1,2], including gels, chewing gum, toothpaste, and films, methods that allow self-evaluation of tooth whitening at home are limited. The two common methods for evaluating the effectiveness of tooth whitening (i.e., tooth shade determination before and after whitening) are visual and digital [3]. Visual methods are subjective, have acceptable precision, and generally use dental shade guides [4,5,6], which are sets of differently colored tooth models called shade tabs, to assess tooth shade. Conversely, digital methods are objective, have higher precision, and use optical devices, such as dental colorimeters [7,8], dental spectrophotometers [9,10], and digital single-lens reflex cameras with color-measuring software [11,12]. However, the cost and lack of portability of these tools make them unsuitable for home use. In addition, they can only be operated by trained technicians. Therefore, tooth shade determination at home by whitening product consumers remains a challenge.

Recent advances in smartphone cameras allow dental photographs to be captured for determining tooth shade [13,14,15,16]. In addition to camera settings such as aperture size, shutter speed, and ISO, other factors that affect tooth shade in smartphone photographs are as follows: (1) illumination conditions, determined by both the intensity of the ambient light and camera flash; (2) tooth appearance conditions, including the region of interest (ROI) for tooth color measurement, determined by the opening of the mouth and tooth size; and (3) tooth position on dental photographs, determined by the positional relationship between the camera and the tooth. Therefore, unifying both illumination and tooth appearance conditions while capturing a dental photograph before and after whitening treatment is required.

Tam et al. [13] reported a smartphone-based shade determination method using dental photographs captured via a rear-facing camera using auto-mode camera settings, a camera-shade tab distance of 14 to 20 cm, no flash, and a light with a color temperature of 4000 K. They used support vector machines (SVM), a computationally efficient technique, for data classification. This SVM-assisted method offers in vitro shade determination with high accuracy, regardless of camera-shade tab distances. Sirintawat et al. [14] reported a shade determination method using a smartphone with an external ring light source and a polarization filter to stabilize illumination conditions, wherein the camera-shade tab distance was fixed at 15 cm. However, smartphone-based methods using rear-facing cameras have been developed for clinical use (e.g., dental restorations) by dentists. Hence, they are not suitable for home use or self-operation via a selfie.

We believe that customers using whitening products must be able to perform tooth shade determination at home using their smartphones. Table 1 presents a comparison between previous studies and this study, which proposes personalized tooth shade determination (pTShaDe). In this study, a smartphone camera application (app) was developed to capture dental photographs in selfie mode under standardized illumination and tooth appearance conditions without additional equipment. Further, the app’s effectiveness in ensuring uniform tooth appearance and its feasibility for self-evaluating the effects of gel whitening were investigated.

## 2. Materials and Methods

### 2.1. Methodology: Design of the App, Developmental Environment, Smartphone, and Camera Settings

This study used an iPhone 11 (Apple Incorporated (Inc.), Cupertino, CA, USA) as the test smartphone. The camera app was developed as an iOS app using Apple’s integrated development environment Xcode (Apple Inc.) with Swift as the programming language. An ambient light sensor was implemented in the pTShaDe system to capture photographs in the dark (0 lx) under standardized illumination conditions. An artificial intelligence technique named facial landmark detection (FLD), which estimates predefined landmarks (eyes, nose, mouth, eyebrows, and face outline) on a facial image and provides geometric information [17], was used to ensure consistent tooth appearance conditions. A class named UIImagePicker in a framework named UIKit was used to control the shot timing of the front-facing camera, and a class named VNDetectFaceLandmarksRequest was used to perform FLD analysis in a framework named Vision, a computer vision framework provided by Apple Inc., to ensure uniform tooth appearance in each photograph. The FLD analysis provides XY coordinate information for 85 landmark points as facial features when a human face is detected in the input image. When the location of the face in the input image cannot be determined, FLD analysis is not performed. Further, the accuracy of FLD analysis is affected by the direction of the face, and the accuracy of a facial image facing the front is higher than that of a face facing sideways.

Dental photographs were captured in the app using auto-mode camera settings with a front-facing flash named Retina Flash. The app assumed a dark room environment to standardize the illumination conditions. Consequently, the two primary functions of uniform illumination and tooth appearance were implemented. For ensuring uniform illumination, an ambient light sensor was used to standardize illumination conditions, realized by utilizing the auto-brightness adaptation function of the iPhone’s display against ambient light. The value of the display brightness was verified to increase linearly from 0.0 to 1.0 with ambient light ranging from 0 to 6.5 lx (Appendix A). In this study, illumination was considered sufficiently low (i.e., dark (0 lx)) when the display brightness value was less than 0.0133 and used as a threshold value. This threshold value was lower than 0.0198 as the upper bound of the 95% confidence interval (CI) at 0 lx (mean = 0.006682, standard error of the mean (SEM) = 0.006682, *n* = 5).

Figure 1 shows the workflow of the developed app. Figure 2 shows representative images for capturing dental photographs using the app (See Appendix A for the dental photograph capturing method). The upper section of the app is designed to capture the first photograph and generate tooth appearance criteria, whereas the lower section is designed to capture subsequent photographs and verify tooth appearance in the captured photographs. After launching the app, the user creates a dark room by adjusting the ambient light in the room (e.g., shutting out sunlight using curtains, etc., and turning off room lighting). Then, the ambient light sensor in the app is initialized by turning display auto-brightness off and on (Settings > Accessibility > Display and Text Size). In the app, the sensor is launched when users activate the front-facing camera. Activation is permitted only if the ambient light is 0 lx; otherwise, the camera is not activated (See Appendix A). After capturing the first photograph (Figure 2a), tooth appearance conditions, determined by the mouth opening, are analyzed via FLD to record tooth appearance conditions in the first photograph (Figure 2b,f) and generate a face image with an outline (Figure 2c). To ensure uniform tooth appearance, a semi-transparent image with (Figure 2d) or without an outline (Figure 2e) is overlaid on the real-time image when users capture subsequent images. Both tooth alignment and image capture are performed by users after complete alignment with the semi-transparent image has been achieved. Tooth appearance criteria are generated from the first photograph using geometric information of the first hexagonal region inside the lips, as analyzed via FLD. The position of the center of gravity and area are used as geometric information for the region. Here, the position of the center of gravity and area of the region in the *n*-th captured photograph are referred to as *G_Hn_* and *A_Hn_*, respectively (*G_H_*_0_ and *A_H_*_0_ are for the first photograph). In the *n*-th capture, the tooth appearance criteria are considered satisfied when *G_Hn_* is located inside the smallest rectangle that can enclose the first hexagonal region (Figure 2g), and *A_Hn_* ranges from 0.7 × *A_H_*_0_ to 1.3 × *A_H_*_0_. Consequently, the captured dental photograph is permitted to be saved. Until the criteria are satisfied, the photograph cannot be saved, and recapturing photographs is urged (See Appendix A). Note that the saved photographs are used as dental photographs to evaluate whitening products, and the first photograph can be freely imported from the photo library on the iPhone for subsequent dental photographs captured using the app.

### 2.2. Tooth Color Measurement and Tooth Appearance Uniformity Evaluation

The photograph obtained using the app was saved in the RGB color space in JPEG format with a size of 2316 × 3088 pixels. A raster graphics editor (Photoshop 2021; Adobe Systems, San Jose, CA, USA) was used to select and crop the upper incisors (Figure 3a,b). The selections were semi-automatically performed using Object Selection in Photoshop to promote reproducibility. After cropping, whiteout parts were removed using MATLAB (MathWorks, Natick, MA, USA). To determine the whiteout parts, the cropped image was temporarily converted to 8-bit grayscale, and white areas above a threshold of 250 were selected as the whiteout parts (Figure 3c). The remaining incisors were marked as the ROI for tooth color measurement in the CIELAB color space (*L**: lightness; *a**: redness-greenness; *b**: yellowness-blueness) [3]. Although spatial color distributions exist within a single tooth [11], the color values over the ROI were measured, and their mean was calculated using the rgb2lab function in MATLAB. Tooth appearance uniformity in the captured dental photographs was evaluated using both geometric and colorimetric information from the upper incisors of seven participants. To examine tooth appearance, the center of gravity observed in the first photograph was designated as the origin *G_I_*_0_ (0, 0) of an orthogonal coordinate system (Figure 3d). The *n*-th centers *G_In_* (*n* = 1 to 5) acquired from five subsequent dental photographs were plotted on the coordinate system. During evaluation, the distance *D* between the origins *G_I0_* and *G_In_* was used to express the gap from the origin directly:

(1)
$$D=\sqrt{{{(x}_{gIn}-x_{gI0})}^{2}+{{(y}_{gIn}-y_{gI0})}^{2}}=\sqrt{{x_{gIn}}^{2}+{y_{gIn}}^{2}}$$
where *x_gIn_* and *y_gIn_* are the coordinates of the center of gravity, *G_In_*.

The incisor areas were also used to evaluate the uniformity of tooth appearance. Normalized area *A* was introduced to express the consistency of tooth appearance:

(2)
$$A=\frac{A_{In}}{A_{I0}}$$
where *A_I_*_0_ is the area of the incisors observed in the first photograph and *A_In_* (*n* = 1 to 5) is the area observed in the subsequent five photographs (Figure 3d).

We also examined the variation in colorimetric information of the upper incisors in the five subsequent photographs captured in succession as an indicator of tooth appearance uniformity. In particular, the coefficients of variation (CV) of incisors, the *L**, *a**, and *b** values, of the subsequent five photographs were used. The center of gravity, area, *L**, *a**, and *b** values of the incisors were measured using MATLAB. In addition, the required time and number of errors per dental photograph were examined to estimate the usability of the app.

### 2.3. Data Analysis

All statistical analyses were performed using a two-sided Student’s *t*-test. In this study, *p* < 0.05 was considered significant.

### 2.4. Feasibility of the App for Personalized Tooth Shade Determination

To verify the feasibility of in vitro tooth shade determination using the application, a shade guide including A1, A2, A3, and A4 tabs (Vintage Halo NCC Standard, Shofu Inc., Kyoto, Japan) [6] was used. For color measurement, a threshold of 240 was used to remove the whiteout parts from the shade tabs in five photographs obtained using the app. The process used to capture the first photograph was used to capture photos of the shade tabs. In this experiment, the feasibility of shade determination using the app, the effects of positional relationships between the camera and tabs, and illumination conditions on color measurement were investigated. To examine the effects of illumination conditions, light with varying illuminance was placed right above the midpoint of the straight line connecting the camera and A2 tab, and its illuminance was varied. To examine the effects of positional relationships, the position of the A2 tab was varied in all three dimensions.

To verify the feasibility of in vivo tooth shade determination, a gel whitening treatment (Opalescence PF Quick 45%; Ultradent Products, Inc., South Jordan, Utah) was conducted after successive pseudo-tooth staining using an instant coffee powder (Blendy Special Taste; Ajinomoto AGF Inc., Tokyo, Japan) and grape juice (Asahi Welch’s Grape 100; Asahi Soft Drinks Co. Ltd., Tokyo, Japan). Similar image processing and color measurement with a tooth appearance uniformity check were performed on the incisors in dental photographs obtained after staining and whitening treatment. The color difference [3] *∆E_ab_* was introduced to express the incisor color differences pre- and post-whitening.

(3)
$$\Delta E_{ab}=\sqrt{{\left(L_{post}^{}-L_{pre}^{}\right)}^{2}+({a_{post}^{}-a_{pre}^{})}^{2}+{\left(b_{post}^{}-b_{pre}^{}\right)}^{2}}$$

where *L_pre_*, *a_pre_*, *b_pre_*, and *L_post_*, *a_post_*, *b_post_* are *L**, *a**, *b** values of incisors before and after staining or whitening, respectively.

Note that in the case of no significant difference between pre- and post-treatment values via a two-sided Student’s *t*-test, a difference of 0 is considered (e.g., if there is no significant difference between *L_post_* and *L_pre_*, 
${\left(L_{post}^{}-L_{pre}^{}\right)}^{2}$
 is 0).

### 2.5. Study Participants

Seven participants, whose ages ranged from 22 to 44 were selected for the experiments to evaluate tooth appearance uniformity in dental photographs obtained by the app. One of them was involved in experiments to verify the feasibility of in vivo tooth shade determination and evaluate the effects of gel whitening after pseudo-staining.

## 3. Results

### 3.1. Evaluation of Tooth Appearance Uniformity in Photographs Obtained Using the App

A significant aspect of the user interface of the developed app is the semi-transparent image displayed while capturing dental photographs. The effects of the semi-transparent images on the uniformity of upper incisor appearance for all seven participants were primarily examined. Figure 4a shows the tooth appearance of a participant (Subject 6) cropped from dental photographs, while the tooth appearances of the other six participants are shown in Appendix A. Figure 4b summarizes the associations between the distances of the centers of gravity of the incisors and two semi-transparent images. When the distance was closer to zero, the incisor positions in the photographs were consistent. One participant (Subject 3) had a significant decrease in distance when a semi-transparent image with an outline was used. No significant difference was observed between the two semi-transparent images in terms of average distance (*n* = 35 for five photographs/participant across seven participants). The normalized area was calculated to evaluate the uniformity of the upper incisor appearance compared to that of the first photograph (Figure 4c). A normalized area closer to 1 signified higher incisor appearance uniformity. One participant (Subject 1) had a normalized area significantly closer to 1 when the semi-transparent image with an outline was used (*n* = 5). Two participants (Subjects 2 and 5) had significantly closer normalized areas when the semi-transparent image without an outline was used. Although there was a significant difference between the two semi-transparent images in the averaged normalized area (*n* = 35), the distance from 1 is comparable (averaged normalized area with outline: 1.055, without: 0.945).

Next, color uniformity using the coefficients of variation for *L**, *a**, and *b**, which are the key factors for accurate tooth shade determination, was investigated (Figure 4d–f). A coefficient of variation closer to zero indicates higher uniformity. Four participants (Subjects 2, 3, 4, and 6) had coefficients of variation of the *L** values closer to 0 when the semi-transparent image with an outline was used. There was no significant difference observed between the two semi-transparent images on the basis of the average coefficient of variation of *L** value for seven participants, with an average of variation for each participant. Four participants (Subjects 1, 2, 5, and 7) had a closer coefficient of variation of *a** when the semi-transparent image without an outline was used. There was no significant difference between the two semi-transparent images on the basis of the average coefficient of variation of the *a** value for seven participants, with an average of variation for each participant. Six participants (Subjects 1, 3, 4, 5, 6, and 7) had a closer coefficient of variation of the *b** value when the semi-transparent image with an outline was used. There was no significant difference between the two semi-transparent images on the basis of the average coefficient of variation of the *b** value for seven participants, with an average of variation for each participant. Table 2 summarizes the association between the two semi-transparent images and the average coefficient of variations of *L**, *a**, and *b**.

To evaluate the usability of the app, the required time and number of errors per photograph were examined. One participant (Participant 7) had a significantly shorter photograph capture time when the semi-transparent image without an outline was used. No significant difference was observed between the two semi-transparent images regarding average photo capture time. One participant (Participant 5) had a significant decrease in the number of errors per capture when the semi-transparent image without an outline was used. No significant difference was observed between the two semi-transparent images regarding the average number of errors.

### 3.2. Feasibility Check of the App for Personalized Tooth Shade Determination

#### 3.2.1. In Vitro Tooth Shade Determination

First, the feasibility of in vitro shade determination using the developed app was examined. To evaluate the reliability of color measurement, shade tab photographs were obtained using the app with a fixed camera-shade tab distance of 20 cm. The measured *L**, *a**, and *b** values are shown in Appendix A. Except for the *L** values of A1 and A2, significant differences among the other combinations were observed. Table 3 summarizes the variability in measurements. The color differences between the tabs are shown in Appendix A. The smallest color difference was observed between A2 and A3 (*∆E_ab_* = 2.8).

In this in vitro experiment, the effect of ambient light illumination on color measurements was investigated (Appendix A). No significant difference in *L**, *a**, and *b** values of the A2 tabs between 0 lx and 3 lx was observed. In addition, the effect of the positional relationship between the camera and A2 tab at 0 lx was investigated by varying the tab positions three-dimensionally with the camera position fixed. The *L**, *a**, and *b** values of the A2 tab were significantly affected by 1 cm variations between the camera and shade tabs (Appendix A). When the camera and tab positions were fixed at 20 cm, the *L** value of the A2 tab was not significantly affected by its horizontal displacement of 8 mm (one-tab horizontal shift), whereas the *a** and *b** values were associated (Appendix A). The *a** value of the tab was not significantly affected by its vertical displacement of 11 mm (one-tab vertical shift), whereas the *L** and *b** values were affected (Appendix A).

#### 3.2.2. In Vivo Tooth Shade Determination

Second, the feasibility of in vivo tooth shade determination was examined. Figure 5 summarizes the results of color measurement before and after gel whitening treatment after successive pseudo-stains with instant coffee powder and grape juice. Significant shade reduction and enhancement in *L** values were observed after grape juice staining and whitening (Figure 5b), respectively. A significant reduction in *a** values was observed after grape juice staining, whereas no significant increase after whitening was observed (Figure 5c). A significant increase and decrease in the *b** value were observed after coffee staining and grape juice staining (Figure 5b), respectively. No significant difference in *b** values was observed after grape juice staining. Table 4 shows color differences between each staining and whitening treatment. Since no significant differences in *L**, *a**, and *b** values were observed between the initial state and post-whitening, the color difference *∆E_ab_* was 0. Against post-whitening, color differences were observed with coffee (*∆E_ab_* = 1.9) and grape juice staining (*∆E_ab_* = 2.5), respectively. Against the initial state, the smallest color difference was observed with coffee staining (*∆E_ab_* = 1.3).

## 4. Discussion

In this study, a novel technique for personalized tooth shade determination called pTShaDe, which utilizes selfie-mode dental photographs obtained via a smartphone to evaluate the results of whitening products, was presented. To realize pTShaDe, an iOS camera app for obtaining dental photographs under standardized illumination and tooth appearance conditions was developed. Although conventional digital methods using traditional tools such as spectrophotometers [9] have higher precision than the proposed method, they have been developed for and used clinically by dentists. Conversely, the proposed method has been developed for consumers using tooth-whitening products at home. To determine precise pre- and post-whitening dental shades, consumers must ensure consistent illumination and tooth appearance conditions. Therefore, the ambient light sensing function checks illumination conditions, and FLD ensures uniform tooth appearance. Since tooth color in photographs obtained by the app depends on both the camera tooth distance and the model of the smartphone (i.e., its camera and flash), the present shade determination method is not an absolute but a relative quantification method.

Our study on tooth appearance uniformity and variability of color measurements in photographs obtained by the app with auto-mode camera settings and retina flash suggests no significant differences between the semi-transparent images with and without an outline in the averaged results (Figure 4b–f). In addition, no significant differences in average time or number of errors per capture have been observed, verifying the usability of the developed app (Figure 4g,h). Therefore, the semi-transparent image type selectable in the app depends on user preference to promote personalization.

From tooth shade determination, reducing variabilities in color measurement, represented as coefficients of variation of *L**, *a**, and *b** values (Table 2), is extremely critical. Variabilities in these coefficients were comparable to those of a previous study that used a smartphone with an external ring light source and a polarized filter [14]. Making the tooth appearance criteria severe is a promising way to achieve this goal. There are two directions for the criteria that can be considered severe. One is narrowing the present acceptable range for both the center of gravity and the hexagonal region (Figure 2g). In particular, an improvement in the criterion for area (from 0.7 × *A_H_*_0_ to 1.3 × *A_H_*_0_ in this study) is more effective because it reflects the distance between the camera and the tooth. The importance of this distance is shown in Appendix A. Another condition is adding one more criterion regarding the shape of the hexagonal region, indicating how to open the mouth, which was not considered in this study. In any case, improving the current criteria is possible; however, making the criteria strict raises a trade-off between the time required and the number of errors per capture. Adding a function to automatically capture photographs when the criteria are met can be effective in managing this trade-off.

Our in vitro study involving shade tabs suggests that the app with auto-mode camera settings and retina flash was stable enough to distinguish the four shade tabs (A1 and A4) for a fixed camera-shade tab distance at 0 lx (Appendix A). This result indicates the highest feasible precision of the proposed method for color measurement (Table 3). Furthermore, the extent of acceptable mismatches regarding illumination conditions for a fixed camera-shade tab distance of 20 cm (0–3 lx) was calculated by this in vitro study (Appendix A). Moreover, the effects of the positional relationship between the camera and shade tab at 0 lx on color measurement were verified (Appendix A). The camera-shade tab distance had the largest impact.

Our in vivo study with pseudo-staining and whitening treatments demonstrated the feasibility of the application of pTShaDe (Figure 5). The smallest color difference *∆E_ab_* of 1.3 observed between the initial state and coffee staining, can be considered the detection limit of the app. Previously, *∆E_ab_* of 1.6 and 1.9 were reported by Ishikawa–Nagai et al. [18] and Thoma et al. [19], respectively. These differences are difficult to perceive with human eyes. The sensitivity for the significant decrease in the *a** value by grape juice staining and the significant increase in the *b** value by coffee staining from the initial value and the detection limit in color difference offer the possibility of the application to monitor tooth stains, allowing us to determine suitable timing for whitening treatment. These results suggest the usefulness of the app for not only evaluating tooth-whitening treatments but also tooth shade determination for direct [20] or indirect [21] restorations.

Our proposed method has limitations in terms of the smartphone to be used, the environment of use, and color measurement. First, as the camera and flash performance directly affect the dental photographs, the same result cannot be obtained with different smartphones. Second, the proposed method requires dark conditions (0 lx) and securing an environment that eliminates the effects of sunlight during the day is necessary. Third, the proposed method requires not one photograph but multiple photographs for color measurements to increase its accuracy. As several variations in tooth appearance are captured in the photographs simultaneously, the proposed method requires multiple photographs to ensure highly accurate color measurements. Fourth, the present method requires an external application (Photoshop and MATLAB) for tooth color measurements. Hence, the integration of a color measurement function in the app is the next step.

## 5. Conclusions

In this study, a novel personalized tooth shade determination method called pTShaDe, which utilizes a smartphone and selfie-mode dental photographs to evaluate the results of whitening products, was presented. To realize pTShaDe, a smartphone camera app was developed to acquire dental photographs under standardized illumination and tooth appearance conditions. The feasibility of the app for pTShaDe was examined by monitoring the *∆Eab* color difference post-gel whitening treatment and pseudo-staining, and the smallest color difference *∆E_ab_* of 1.3 was confirmed as the detection limit of the app. However, pTShaDe remains a relative quantification method and has the aforementioned limitations; this study will serve as the basis for an evidence-based selection of tooth-whitening products.

## Figures and Tables

**Figure 1 diagnostics-13-01969-f001:**
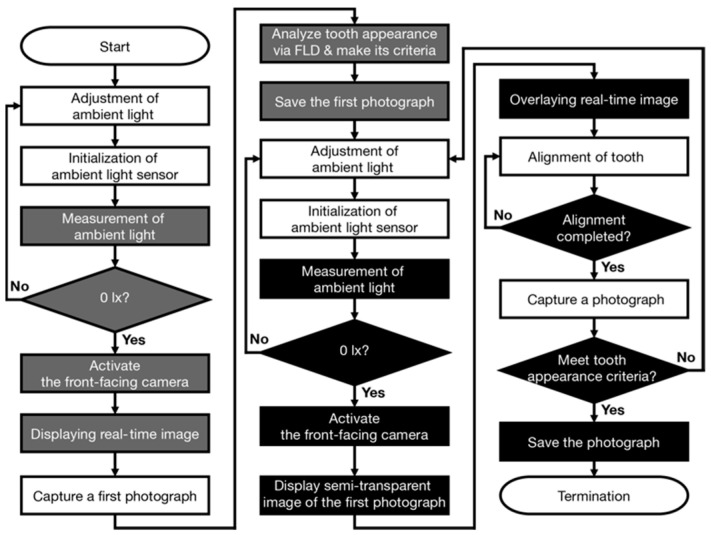
Flow chart of the developed app to acquire dental photographs for the personalized tooth shade determination. White-colored tasks are performed by the app’s users (□). Gray-colored tasks for capturing the first photograph to meet both the semi-transparent image and tooth appearance criteria are carried out by the app (■). Black-colored tasks for capturing photographs to evaluate whitening products (i.e., pre- and post-whitening photographs) are carried out by the app (■).

**Figure 2 diagnostics-13-01969-f002:**
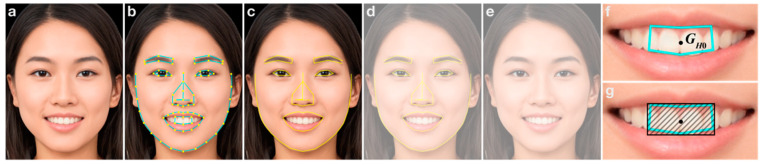
Representative images for capturing dental photographs using the app. (**a**) The first photograph captured. (**b**) Image after FLD analysis. (**c**) Image removed the hexagonal line inside the lips in (**b**). (**d**) Semi-transparent image with and (**e**) without outline displayed in the app during subsequent capturing. (**f**) Magnified image of the mouth from (**b**). GH0 denotes the center of gravity of the hexagonal region. (**g**) Tooth appearance criterion related to the position of the center of gravity of the hexagonal region for subsequent capture. The hatched area is the smallest rectangle that can enclose the first hexagonal region. Photos of faces are used with permission by Generated Photos (https://generated.photos (accessed on 29 March 2023)).

**Figure 3 diagnostics-13-01969-f003:**
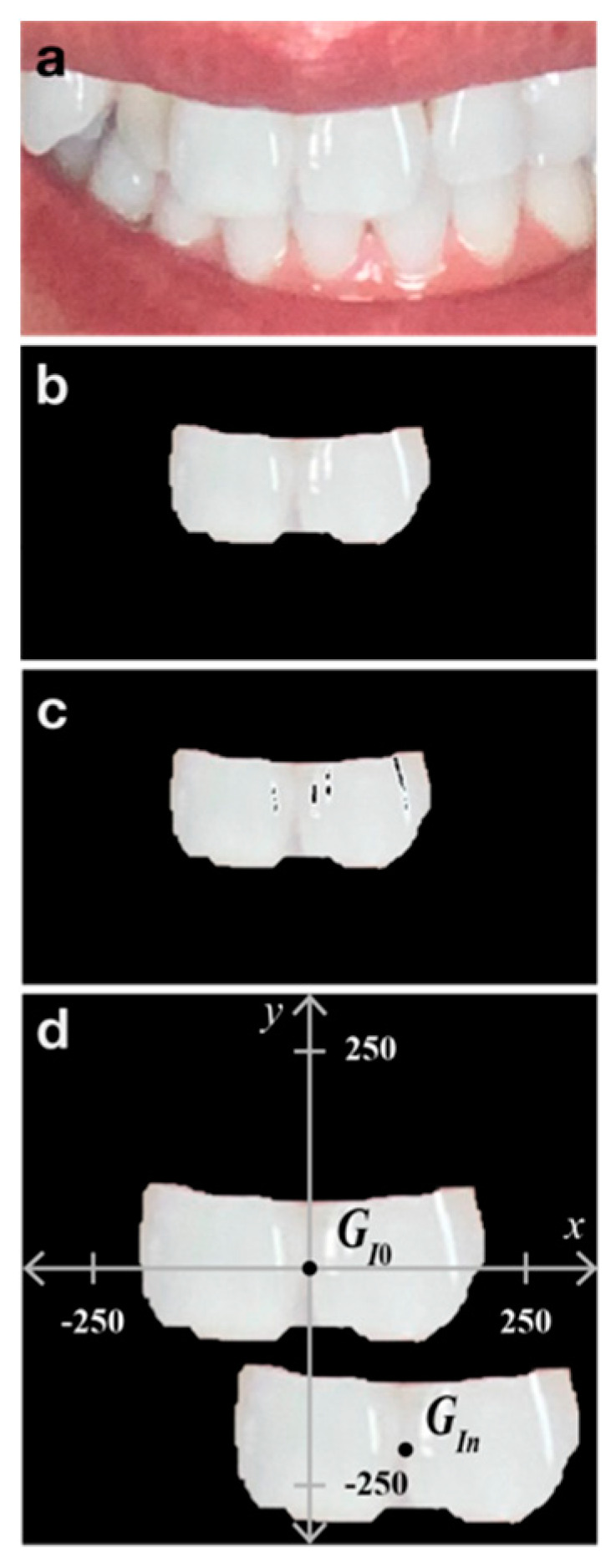
Representative images for tooth color measurement and evaluation of tooth appearance uniformity. (**a**) The first captured image. (**b**) Cropping the area of the incisors. (**c**) Whiteout-removed image. (**d**) Orthogonal coordinate system with the centers of gravity of the incisors. The origin is designated using the centers of gravity of the incisors observed in the first photograph.

**Figure 4 diagnostics-13-01969-f004:**
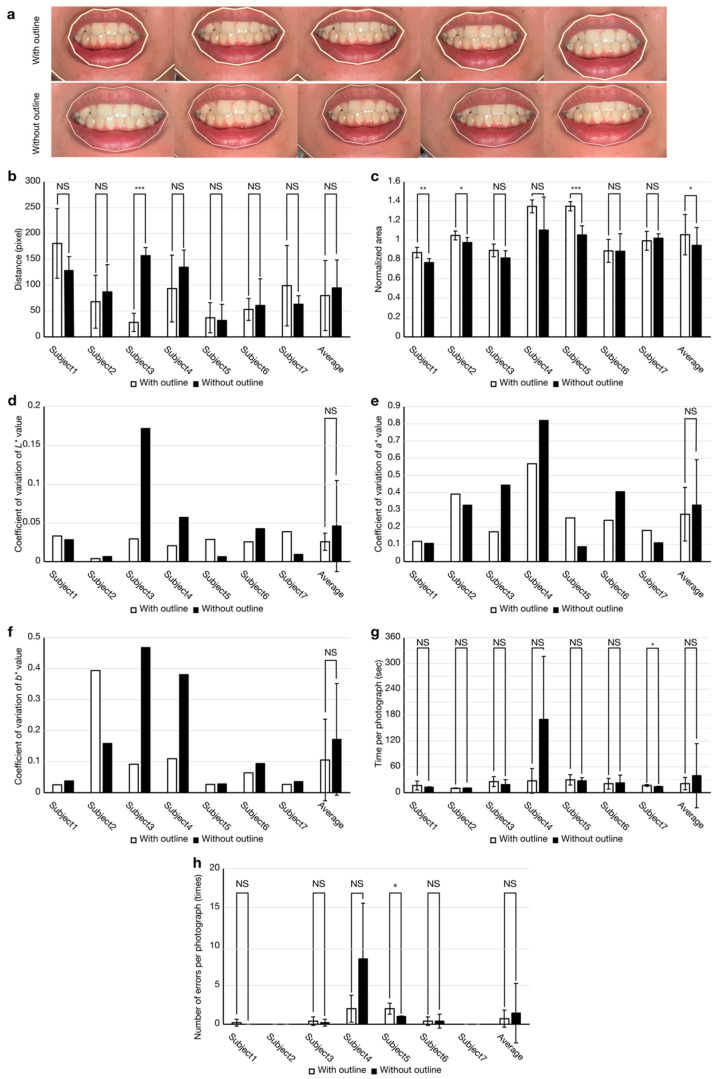
Tooth appearance uniformities in the captured dental photographs and app usability. (**a**) Cropped images of mouth from dental photographs of a participant. (**b**) Distance between the center of gravity of the incisors in the first photograph and five subsequent photographs. (**c**) Areas normalized by incisors obtained in the first captured photograph. (**d**–**f**) Coefficient of variations of *L**, *a**, and *b**. (**g**) Required time and (**h**) Number of errors per photograph (NS, not significant; ∗ *p* < 0.05, ∗∗ *p* < 0.01, ∗∗∗ *p* < 0.001; *n* = 5 at each participant in (**b**,**c**,**g**,**h**). *n* = 35 at average in (**b**,**c**,**g**,**h**). *n* = 7 at average in (**d**–**f**)). Error bars indicate ± standard deviation (SD).

**Figure 5 diagnostics-13-01969-f005:**
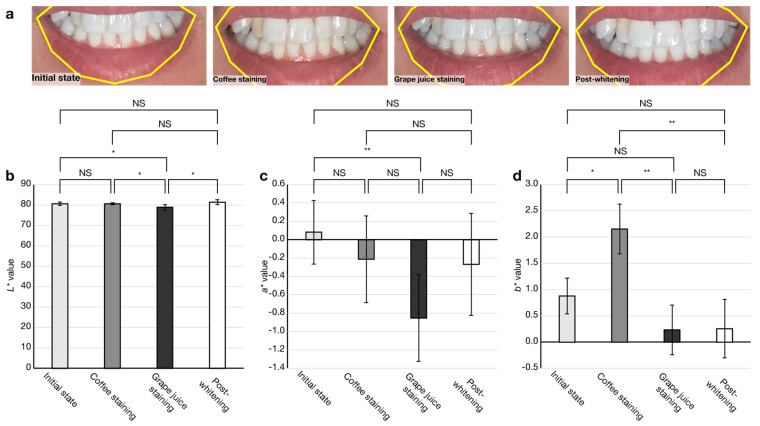
Feasibility check of the app for tooth shade determination. (**a**) Representative images of pre- and post-gel whitening treatment with intermediate pseudo-staining. Color measurement results of (**b**) *L** value, (**c**) *a** value and (**d**) *b** value. (NS, not significant; ∗ *p* < 0.05, ∗∗ *p* < 0.01; *n* = 5). Error bars indicate ± SD.

**Table 1 diagnostics-13-01969-t001:** Comparison between previous studies and this study.

Authors	Smartphone	External Light Source	Filter	Capture Mode	Potential Target Application and Its User
Tam et al. [13]	iPhone 6 Plus	None	None	Portrait	Clinical use by dentists
Sirintawat et al. [14]	Galaxy Note 20 Ultra 5G	Light correcting device	Polarization	Portrait	Clinical use by dentists
Mohammadi et al. [15]	iPhone 7	Light correcting device	Polarization	Portrait	Clinical use by dentists
Jorquera et al. [16]	iPhone XS	Light correcting device	Polarization	Portrait	Clinical use by dentists
Kusayanagi et al. [current study]	iPhone 11	None	None	Selfie	Home use by whitening product customers

**Table 2 diagnostics-13-01969-t002:** Variability of color measurements of photographs captured by the developed app.

				95% CI	
	Group	Average ^1^	SEM	Lower Bound	Upper Bound
CV of *L** value	With	0.0256	0.0042	0.0173	0.0338
	Without	0.0459	0.0222	0.0023	0.0894
CV of *a** value	With	0.2748	0.0588	0.1596	0.3899
	Without	0.3279	0.0996	0.1327	0.5230
CV of *b** value	With	0.1053	0.0497	0.0078	0.2028
	Without	0.1717	0.0681	0.0383	0.3051

^1^ *n* = 7.

**Table 3 diagnostics-13-01969-t003:** Variability in color measurement using shade tabs with a fixed camera tab distance. The color measurements from five photographs of each tab (A1~A4) were merged.

			95% CI	
	Average ^1^	SEM	Lower Bound	Upper Bound
CV of *L** value	0.0052	0.0006	0.0040	0.0063
CV of *a** value	0.0322	0.0028	0.0268	0.0376
CV of *b** value	0.0116	0.0035	0.0048	0.0048

^1^ *n* = 4.

**Table 4 diagnostics-13-01969-t004:** Color difference between pre- and post-gel whitening after pseudo-staining.

*ΔE_ab_* ^1^	Initial State	Coffee Staining	Grape Juice Staining	Post-Whitening
Initial state	−	1.3	2.0	0
Coffee staining	1.3	−	2.6	1.9
Grape juice staining	2.0	2.6	−	2.5
Post-whitening	0	1.9	2.5	−

^1^ *n* = 5.

## Data Availability

The data presented in this study are available on request from the corresponding authors.

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
