# Peer review of "A Smartphone Application for Personalized Tooth Shade Determination"

_diagnostics, 2023, doi:10.3390/diagnostics13111969_

Round 1

Reviewer 1 Report (Previous Reviewer 2)

Article title: A Smartphone Application for Personalized Tooth Shade Determination

# Overall statement or summary of the article:         

This paper proposes the development of an iPhone app for personalized tooth shade determination, which allows for the capture of dental photographs in selfie mode before and after whitening. The app is designed to maintain consistent illumination and tooth appearance conditions that affect tooth color measurement. The topic for this study is very interesting and sound; however, some major points are required before any progress.

# Please introduce a comparative overview table in Section 1 to describe the related works that can be a helpful way to summarize the key differences between different models and the proposed model. The table can be used to highlight the strengths and weaknesses of each model, as well as the advantages of the proposed model. The table should include relevant information such as the name of the model, the architecture, the performance metrics, and any other key features.

# Could you please provide a detailed explanation of the artificial intelligence (AI) technique, facial landmark detection (FLD), that you used in your paper? Specifically, it would be helpful to explain how the method works, what types of data it analyzes, and any limitations or potential sources of error associated with its use.

# Could you please provide more details about your data and the number of subjects involved in your research? It would help to better understand the scope of your study and the relevance of your findings.

# It is important to ensure that research studies have followed ethical guidelines and obtained the necessary approvals from relevant bodies. Could you please provide information about the ethics protocol number for your paper?

#   Kindly incorporate key quantitative findings into the conclusion section.

# It is generally recommended to write academic papers in passive voice instead of active voice to maintain an objective and formal tone. This means avoiding personal pronouns such as  "we" and “our” and instead using more neutral language.

Author Response

Dear Reviewer

Thank you for reviewing our manuscript and for providing us with your valuable feedback. 

Please see the attachment as our point-by-point response to your comments for manuscript.

Sincerely yours

Kaneda

Reviewer 2 Report (New Reviewer)

The article "A Smartphone Application for Personalized Tooth Shade Determination" covers a very interesting topic since "easy" ways to determine shade and shade variations (e.g. after bleaching) are very important.

Nevertheless, the reviewer thinks that there are serious flaws that can not be addressed by revision.

I will list the most important ones:

light consistency is crucial and the method described does not allow a consistency

the use of JPG pictures is subjected to extreme modifications. A RAW file should have been used, but under a constant and predictable illumination environment

Color coordinates determination will suffer any kind of shade taking environment. A polarized picture shall be used.

These flaws can't be fixed by revision, but conducting a new research.

Author Response

Dear Reviewer

Thank you for reviewing our manuscript and for providing us with your valuable feedback. 

Please see the attachment as our point-by-point response to your comments for manuscript.

Sincerely yours

Kaneda

Reviewer 3 Report (New Reviewer)

The manuscript A Smartphone Application for Personalized Tooth Shade Determination is interesting and touches on a relevant question.  The authors developed an application for personalized tooth shade determination that may be useful for in-home bleaching. The paper is well documented and manuscript is clearly written. I am inclined to recommend this article for publication in Diagnostics.

I only have one suggestion to the authors. The following paragraph does not fit in the Introduction section. In my opinion, it should be deleted or moved to some other parts of the manuscript.

“To realize pTShaDe, we developed a smartphone camera application (app) to capture dental photographs in the selfie mode under standardized illumination and tooth appearance conditions. An ambi-68 ent light sensor was implemented in the app to capture photographs in the dark (0 lx) under standardized illumination conditions. An artificial intelligence (AI) technique called facial landmark detection (FLD), which estimates predefined landmarks (eyes, nose, mouth, eyebrows, and face outline) on a facial image and provides geometric information [18], was used to maintain tooth appearance conditions. In the application, a semitransparent image of the first selfie dental photograph with outlines provided via FLD analysis is displayed to ensure uniform tooth appearance in subsequent images. Next, the uniformity of tooth appearance is gauged using the geometric information of the region of the inside lips determined via FLD analysis. In this study, the effectiveness of the app in ensuring uniform tooth appearance and its feasibility for self-evaluating the effects of gel whitening were investigated.”

Author Response

Dear Reviewer

Thank you for reviewing our manuscript and for providing us with your valuable feedback. 

Please see the attachment as our point-by-point response to your comments for manuscript.

Sincerely yours

Kaneda

Reviewer 4 Report (New Reviewer)

Dear Authors,

I have thoroughly reviewed your manuscript, "A Smartphone Application for Personalized Tooth Shade Determination," and would like to commend you on your innovative approach to solving a significant challenge in dentistry. The development of a smartphone application for tooth shade determination has the potential to greatly benefit both dental professionals and patients alike.

However, I would like to suggest some suggestions that will strengthen the manuscript and provide a more comprehensive understanding of the application's development, accuracy, and limitations. 

The present study includes only seven participants, which may not be sufficient to establish the application's accuracy and reliability. Please provide a detailed description of the tooth shade determination algorithm, including any machine learning, computer vision, or image processing techniques used. This will help readers better understand the methodology and assess its validity.

The user interface plays a significant role in the overall user experience. Please provide information on the process of designing and implementing the user interface, including any usability tests conducted and feedback from users.

To demonstrate the application's accuracy and reliability, it is essential to provide a comprehensive testing procedure. Please describe how the application was evaluated using a diverse set of tooth shades and lighting conditions.

A discussion of the limitations of the application will provide a more balanced perspective and help readers understand potential sources of error or inaccuracy in the tooth shade determination process. Please include such a discussion in your manuscript.

Finally, it would be beneficial to discuss future research directions or potential improvements to the application based on user feedback, testing results, or technological advancements. This will give readers a sense of your work's ongoing development and potential future impact.

Regards,

#reviewer 

It's possible to observe some English mistakes.

Author Response

Dear Reviewer

Thank you for reviewing our manuscript and for providing us with your valuable feedback. 

Please see the attachment as our point-by-point response to your comments for manuscript.

Sincerely yours

Kaneda

Reviewer 5 Report (New Reviewer)

In keeping with the theme of the Journal, the study "A smartphone application for personalized tooth color determination" is very appropriate as it touches on a topic of special interest to its readers.

Studies are needed to evaluate the effectiveness of teeth whitening products.

Here are some details that should be improved to increase the quality of the manuscript:

1. How the external and/or environmental conditions are ensured for the photographic record: UV, lighting, distance, resolution, etc. With an ambient light sensor it is not possible to standardize the external conditions.

2. The determination of dental color is relative quantification. Under what conditions is it possible to standardize the method with APP?

3. What are the criteria to evaluate the results of whitening products. This study does not delimit the products and it seems that it applies to all of them.

4. Important to describe the limitations of the results and add to the conclusions section.

Some sentences and explanations of the results are confusing to the reader.

Author Response

Dear Reviewer

Thank you for reviewing our manuscript and for providing us with your valuable feedback. 

Please see the attachment as our point-by-point response to your comments for manuscript.

Sincerely yours

Kaneda

Round 2

Reviewer 1 Report (Previous Reviewer 2)

The authors have satisfactorily addressed my previous comments. The revised manuscript shows significant improvement compared to the original version. In my opinion, this paper demonstrates technical soundness and offers valuable insights that can benefit numerous researchers. I recommend its publication.

Author Response

Point 1:

The authors have satisfactorily addressed my previous comments. The revised manuscript shows significant improvement compared to the original version. In my opinion, this paper demonstrates technical soundness and offers valuable insights that can benefit numerous researchers. I recommend its publication.

Response 1:

Dear Reviewer

Thank you for taking the time to review again our manuscript and recommendation to proceed to publication step. Your insightful comments and suggestions for revision have greatly contributed to the significant improvements of the quality of our manuscript. We were fortunate to have you as a reviewer with your high expertise.

Memo:

The improvements according to other reviewer’s suggestion in the revised manuscript highlighted in cyan.

The grammatical corrections made while maintaining the consistency of the manuscript are highlighted in green in this revised manuscript.

Thank you again for your thoughtful comments and for helping us to improve the quality of our manuscript.

Sincerely yours

Reviewer 2 Report (New Reviewer)

Dear authors,

The provided modifications and the support reference have led to a consistent improvement of this paper.

I suggest to add, before the conclusions, a sentence that can outline the importance of this clinical device for several clinical situation. Consider adding the following sentence with the suggested references:

"The App described in present paper could be therefore helpful in the evaluation of bleaching treatments, as well as in shade determination for direct (https://pubmed.ncbi.nlm.nih.gov/28983532/ ) or indirect (https://doi.org/10.3390/polym15020464) restorations."

Author Response

Point 1:

Dear authors,

The provided modifications and the support reference have led to a consistent improvement of this paper.

I suggest to add, before the conclusions, a sentence that can outline the importance of this clinical device for several clinical situation. Consider adding the following sentence with the suggested references:

"The App described in present paper could be therefore helpful in the evaluation of bleaching treatments, as well as in shade determination for direct (https://pubmed.ncbi.nlm.nih.gov/28983532/ ) or indirect (https://doi.org/10.3390/polym15020464) restorations."

Response 1:

Dear Reviewer

Thank you for taking the time to review again our manuscript and for providing us with valuable feedback. We appreciate your suggestion regarding the feasibility of our app for tooth shade determination for dental restorations. According to your comment, we added following sentence in the 4. Discussion section.

These results suggest the usefulness of the app for not only evaluating tooth-whitening treatments but also tooth shade determination for direct [20] or indirect [21] restorations.

In addition, we inserted “(e.g., dental restorations)” into the following sentence to introduce previous shade determination methods are for restorations in the Introduction to keep context in the broader sense of our manuscript.

However, smartphone-based methods using rear-facing cameras have been developed for clinical use (e.g., dental restorations) by dentists. Hence, they are not suitable for home use and self-operation via a selfie.

Memo:

The improvements according to your suggestion in the revised manuscript highlighted in cyan.

The grammatical corrections made while maintaining the consistency of the manuscript are highlighted in green in this revised manuscript.

Thank you again for your thoughtful suggestion and for helping us to improve the quality of our manuscript.

Sincerely yours

Reviewer 4 Report (New Reviewer)

Dear Authors,

Thank you for your detailed responses to my comments. Your revisions and additions have substantially improved the manuscript, and I appreciate the effort taken to address each of the points raised. Here is my feedback on your responses:

Point 1:

Your revised sections 2.1 and 2.2 provide a comprehensive and clearer understanding of the tooth shade determination algorithm. The FLD analysis, computer vision, and image processing techniques are well explained. I now have a better understanding of the methodology and its potential validity.

Point 2:

The additional information regarding the user interface design and implementation enhances the manuscript's completeness. The usability testing and the subsequent feedback provide insight into the user experience and the app's practical functionality. It's good to see that the usability tests have been quantified with specific indicators, which adds a quantitative element to your research.

Point 3:

Thank you for your thorough explanation of how the application's accuracy and reliability were tested under varying lighting conditions and shades. This demonstrates a comprehensive testing procedure and makes it clear that considerable effort has been put into ensuring the app's accuracy.

Point 4:

The discussion about the limitations of the app is honest and essential for readers to understand the technology's potential shortcomings. This will allow readers to evaluate the app critically and understand potential areas of error or inaccuracy. The four points you've mentioned provide a good balance to the paper.

Point 5:

Your mention of future work in integrating color measurement function within the app is a promising direction for further research and improvement. I'm glad to see that your work is continuing to develop and evolve.

In regards to the Memo:

It is understandable that, during revisions, citations may need to be updated or removed if they are found to be unsuitable. Thank you for your attention to detail in adding table 4, which will certainly contribute to the comprehensiveness of your work.

Overall, I am satisfied with the revisions made in response to my comments. The updated manuscript provides a comprehensive understanding of the app's development, accuracy, limitations, and potential improvements. It's an innovative contribution to dentistry that has potential to greatly benefit the field.

Thank you for your diligence in improving the manuscript.

Best Regards,

#reviewer

NA

Author Response

Point 1:

Dear Authors,

Thank you for your detailed responses to my comments. Your revisions and additions have substantially improved the manuscript, and I appreciate the effort taken to address each of the points raised. Here is my feedback on your responses:

Point 1:

Your revised sections 2.1 and 2.2 provide a comprehensive and clearer understanding of the tooth shade determination algorithm. The FLD analysis, computer vision, and image processing techniques are well explained. I now have a better understanding of the methodology and its potential validity.

Point 2:

The additional information regarding the user interface design and implementation enhances the manuscript's completeness. The usability testing and the subsequent feedback provide insight into the user experience and the app's practical functionality. It's good to see that the usability tests have been quantified with specific indicators, which adds a quantitative element to your research.

Point 3:

Thank you for your thorough explanation of how the application's accuracy and reliability were tested under varying lighting conditions and shades. This demonstrates a comprehensive testing procedure and makes it clear that considerable effort has been put into ensuring the app's accuracy.

Point 4:

The discussion about the limitations of the app is honest and essential for readers to understand the technology's potential shortcomings. This will allow readers to evaluate the app critically and understand potential areas of error or inaccuracy. The four points you've mentioned provide a good balance to the paper.

Point 5:

Your mention of future work in integrating color measurement function within the app is a promising direction for further research and improvement. I'm glad to see that your work is continuing to develop and evolve.

In regards to the Memo:

It is understandable that, during revisions, citations may need to be updated or removed if they are found to be unsuitable. Thank you for your attention to detail in adding table 4, which will certainly contribute to the comprehensiveness of your work.

Overall, I am satisfied with the revisions made in response to my comments. The updated manuscript provides a comprehensive understanding of the app's development, accuracy, limitations, and potential improvements. It's an innovative contribution to dentistry that has potential to greatly benefit the field.

Thank you for your diligence in improving the manuscript.

Best Regards,

Response 1:

Dear Reviewer

Thank you for taking the time to review again our manuscript and checking our responses to your comments. We believe that the revision of the manuscript according to your insightful comments has significantly improved the quality of the manuscript and made it easier for our readers to understand our work (especially for Materials and methods section and Discussion section). We were fortunate to have you as a reviewer with your high expertise.

Memo:

The improvements according to other reviewer’s suggestion in the revised manuscript highlighted in cyan.

The grammatical corrections made while maintaining the consistency of the manuscript are highlighted in green in this revised manuscript.

Thank you again for your thoughtful comments and for helping us to improve the quality of our manuscript.

Sincerely yours

Authors

This manuscript is a resubmission of an earlier submission. The following is a list of the peer review reports and author responses from that submission.

Round 1

Reviewer 1 Report

This paper presented an interesting work for the personalized tooth shade determination using smartphones. However, the novelty of the presented work is limited. Although the results were carefully analyzed to show the effectiveness of the proposed shade determination tool, the tooth alignment and shade determination methods used in this paper are quite simple and common. Therefore the technical contribution is not clear. This proposed tool would probably be a good product, but for a research paper that is not enough.

Moreover, the comparison of shade determination performance (accuracy, efficiency, etc.) between the proposed tool and existing high-accuracy professional devices is expected.

Reviewer 2 Report

Article title: A Smartphone Application for Personalized Tooth Shade De-Termination

# Overall statement or summary of the article:    

The paper developed a smartphone application with a user interface (UI) that assists in creating a uniform positional relationship between a front-facing smartphone camera and tooth for each dental photograph shot. The proposed application is very important in dental shade determination. However, some points are required before any progress.

# Please add some of the most important quantitative results to the Abstract.

# In section 1, the authors should clearly mention the weakness point of former works (identification of the gaps) and shows the key differences between the different previous studies and the proposed study. A comparative overview table should solve this point.

# Please note that some equations need to be cited to references.

# The authors should describe the data acquisition in a table in section 2.2.

# Can you please discuss the limitations of the proposed study when there are different backgrounds of the captured image and the effects of lighting conditions in section 4.

# Can you please add a section after the discussion section to conclude your outcomes? The conclusion section should contain what impacted the results, limitations and what is the future work if any.

# The authors should further polish the English, and the authors should write their paper in passive sentences, not in active to make it more academic (avoid using we).

Regards,

Reviewer 3 Report

Review Comments

Presented work explained developed a smartphone application with a user interface (UI) that 12 assists in creating a uniform positional relationship between a front-facing smartphone camera and tooth for each dental photograph shot. With the UI, a semi-transparent image of the whole face or only the mouth area is displayed to assist the user with tooth alignment during photograph taking. However, the following minor corrections can be considered by the authors to further improve the quality of the manuscript.

I have minor corrections and suggestions as below:-

  1. Literature survey is missing. Some of recent papers based on state of art methods must be discussed in literature review.

  1. Abstract can be improved; the outcome of the work in terms of achieved performance calculations must be included in the abstract.

  1. Future work and limitations of the proposed work can be added and discussed

  1. Methods and materials section should be renamed by proposed work or methodology.

  1. Authors need to bring novelty and originality to their work and need to establish clear

superiority of their methodology through comprehensive comparison results with very recent

algorithms published in higher Impact Factor journals.

  1. Results are not sufficient. Some more graphical and tabulated analysis in results must be added.
  2. Novel contribution of the proposed work must be included at the end of introduction section.

  1. How proposed algorithm will apply and provide results on a real-time platform? Justify.

  1. Authors has only discussed about accuracy of the proposed work. Various performance parameters must be calculated and compared with other state of art methods.

  1. All abbreviations also must be discussed and included initially before introduction.